# An End-to-End Dynamic Posture Perception Method for Soft Actuators Based on Distributed Thin Flexible Porous Piezoresistive Sensors

**DOI:** 10.3390/s23136189

**Published:** 2023-07-06

**Authors:** Jing Shu, Junming Wang, Kenneth Chik-Chi Cheng, Ling-Fung Yeung, Zheng Li, Raymond Kai-yu Tong

**Affiliations:** 1Department of Biomedical Engineering, The Chinese University of Hong Kong, Hong Kong SAR 999077, China; 1155138492@link.cuhk.edu.hk (J.S.); 1155151867@link.cuhk.edu.hk (J.W.); lfyeung@cuhk.edu.hk (L.-F.Y.); 2Department of Biomedical Engineering, The Hong Kong Polytechnic University, Hong Kong SAR 999077, China; kenneth-c.cheng@polyu.edu.hk; 3Research Institute for Sports Science and Technology, The Hong Kong Polytechnic University, Hong Kong SAR 999077, China; 4Department of Surgery, The Chinese University of Hong Kong, Hong Kong SAR 999077, China

**Keywords:** soft robotics, soft sensors, flexible porous structures, deep learning, long short-term memory

## Abstract

This paper proposes a method for accurate 3D posture sensing of the soft actuators, which could be applied to the closed-loop control of soft robots. To achieve this, the method employs an array of miniaturized sponge resistive materials along the soft actuator, which uses long short-term memory (LSTM) neural networks to solve the end-to-end 3D posture for the soft actuators. The method takes into account the hysteresis of the soft robot and non-linear sensing signals from the flexible bending sensors. The proposed approach uses a flexible bending sensor made from a thin layer of conductive sponge material designed for posture sensing. The LSTM network is used to model the posture of the soft actuator. The effectiveness of the method has been demonstrated on a finger-size 3 degree of freedom (DOF) pneumatic bellow-shaped actuator, with nine flexible sponge resistive sensors placed on the soft actuator’s outer surface. The sensor-characterizing results show that the maximum bending torque of the sensor installed on the actuator is 4.7 Nm, which has an insignificant impact on the actuator motion based on the working space test of the actuator. Moreover, the sensors exhibit a relatively low error rate in predicting the actuator tip position, with error percentages of 0.37%, 2.38%, and 1.58% along the x-, y-, and z-axes, respectively. This work is expected to contribute to the advancement of soft robot dynamic posture perception by using thin sponge sensors and LSTM or other machine learning methods for control.

## 1. Introduction

Soft robots, a novel category of robots that has emerged in recent decades, are characterized by their high flexibility, compliance, and safety [1]. Compared with conventional rigid robotics, soft actuators and robotic devices have advantages in human–robot interaction, range of motions, fabrication difficulties, and adaptations to varying environments [2]. However, the perception of posture, which can be easily represented by the status of joints and links in rigid-link robots, presents a significant challenge in the development of soft robotics due to the difficulty in accurately modeling elastic continuous structures.

Researchers have developed sensing methods, which can be broadly categorized into two main groups: on-board sensing and external sensing of robot posture. On-board sensors, such as flexible bending gauges [3,4] and inertial measurement units (IMUs) [5,6], may lack flexibility and restrict the robot’s motion. Fully soft sensors embedded in flexible materials [7,8,9,10] require complex fabrication methods or specialized equipment. To overcome these challenges, researchers have also utilized external sensing methods, such as vision-based sensing [11,12,13] and magnetic tracking [14,15,16], to obtain posture information for soft robots, particularly for proof-of-concept use in laboratory environments.

As a result, there is an increasing need to develop advanced and flexible sensing techniques that can accurately detect the postures of soft robots. In [17,18], researchers developed a piezoresistive actuator body by mixing carbon fiber with elastomer, resulting in changes in resistance values during deformation. In [19], researchers achieved soft robot posture sensing by utilizing a kirigami-inspired sensor and a deep learning calibration method, representing an innovative approach to the closed-loop control of soft robots. In the recent work of Ye et al. [20], a significant breakthrough has been achieved in robotics by utilizing a kirigami-inspired porous sensor for robot teleoperation. In addition to their use in detecting robot postures, flexible sensing techniques have found extensive applications in other areas that require safe and comfortable human–robot interactions, such as the development of artificial skins [21,22] and the monitoring of human body conditions and motions [23,24,25]. In this paper, we propose an end-to-end method that effectively captures the dynamic behavior of soft actuators, with potential applications for perceiving the posture and dynamic behavior of objects during deformation, such as bellow-shaped soft actuators, where deformation is mainly driven by buckling [26]. Specifically, the neighbor wall angle α (see Figure 1) changes during actuator deformation, allowing us to indirectly perceive the posture of the soft robot through the observation of angle changes.

In this study, we utilized commercially available thin conductive sponges as the sensor material, which were attached to the lateral surface of a three-degree-of-freedom (DoF) soft actuator. Previous research has demonstrated that the piezoresistive properties of thin conductive sponges enable the detection of deformation and tactile information of attached objects [22,27,28,29,30,31,32,33]. Due to their flexible and lightweight nature, sensors developed with sponge materials have been widely employed in human–robot interactions, including tactile sensing [22,27,29,30] and human motion detection [28]. The porous and flexible nature of the conductive sponge structure minimizes impediments to the movement of attached objects induced by sensor deformation.

Although recent research work has explored the feasibility of collecting the tactile information of robots using conductive sponge sensors [31,34,35], only a few studies have employed sensors developed with conductive sponge material to detect or estimate the deformation of soft robots. In our previous work [36], we proposed a kirigami-inspired flexible sponge sensor that could estimate the bending angle of a single degree-of-freedom flexible fiber-reinforced bending actuator. By connecting the sensor information and calibration neural network organically, we achieved highly accurate robot posture estimation. In this paper, we simplified the sensor structure to make it suitable for bending detection and increased the sensor number, enabling us to achieve dynamic 3D posture perception of soft robots. To the best of our knowledge, this is the first study to seamlessly combine sponge sensors and a neural network calibration method to achieve soft robot dynamic 3D posture perception.

We employed an end-to-end method in our study to calibrate our sensing system via RNN deep learning. This approach obviates the need for developing a complex kinematic model. The idea has been widely explored and validated in the soft robotics community [9,19,36,37,38,39] as a means of addressing the non-linearity and hysteresis issues associated with data. We utilized a long short-term memory (LSTM) neural network [40], which is a specialized type of recurrent neural network (RNN) that can handle sequential data with the aforementioned difficulties. We trained an LSTM neural network using feedback sensing signals from nine sensors distributed on the side wall of the actuator. The posture of a 3-DoF soft actuator was predicted using this network in conjunction with the input sensor value, and the overall root mean square error (RMSE) value was calculated to gauge the accuracy of our sensing system. Despite the acceptable prediction accuracy of the sensor, certain issues were uncovered during the study. The primary issue concerns the saturation of sensor readings when the sensor approaches full compression, thereby leading to a loss of prediction accuracy in situations where the attached object undergoes extreme deformation, which would be indicated in the experimental results of sensor characterization.

Overall, this paper makes the following contributions:(1)An end-to-end (indirect) method for soft robot posture perception with flexible porous piezoresistive sensors via the RNN deep learning approach.(2)The design and characterization of a scalable miniature flexible bending sensor that can be quickly attached to existing bellow-shaped actuators to perceive the posture and dynamic behaviors as sensing units while minimizing the effect of motion restriction.(3)The validation of our proposed method on a small-scale 3 DoF bellow-shaped soft robot actuator with three pneumatic chambers.

## 2. Design Overview and Rationale

### 2.1. Soft Robot and Flexible Porous Piezoresistive Sensor Design

Fluidic elastomer actuators (FEAs) are widely used in soft robotics. To perceive the posture of FEAs, researchers have employed various methods, including flexible bending gauges [3], sensors with micro-channels and liquid metal [9], piezoresistive materials embedded in the actuator body [41], fiber optic intensity-based sensors [42], and others. As mentioned in Section 1, these sensing methods can either restrict the motion of soft actuators or require bulky serving systems. Consequently, researchers have pursued a sensing method that can integrate independent sensors into the FEA body. Such a method was first developed in [19], which bypassed the need for designing both the soft actuators and their sensing systems. In this study, we present a robot sensing system that utilizes a similar distributed sensing method, employing bending-based thin conductive sponge sensors. Our approach can be easily applied to the surface of soft actuators that undergo deformation during changes in posture.

We designed and fabricated a negative pressure-powered bellow-shaped FEA. The FEA body was fabricated using casting techniques and silicon material (Smooth-Sil 945, Smooth-On Inc., East Texas, PA, USA). The actuator comprises three hexagonal bellow-shaped chambers, with hex-edge lengths of 5 mm, bonded in parallel using silicon adhesive (Sil-Poxy, Smooth-On Inc., USA). The angle between two bellow walls is 55 degrees, and each chamber was constructed by bonding two half-chambers and two sealing caps with the same silicone adhesive. The fabrication process of the FEA is illustrated in Figure 2c–e.

The planar fabrication method has been applied in the sensor fabrication, similar to the methods used in [36,43]. By employing such manufacturing techniques, it is possible to produce the sensors in small batches. The sketches of the planer fabrication process are shown in Figure 2a. Each flexible bending sensor comprises two layers fabricated using planar fabrication methods and cut by a laser cutter. The top layer is made of a 1 mm thin layer and 2 mm wide PU conductive sponge material (Beilong Inc., Wujiang, China) with a constant thickness, while the bottom layer is composed of adhesive (300LSE, 3M, Saint Paul, MI, USA). Copper electrodes with leading wires were attached to the two ends of the sponge using silver conductive adhesives (3703, Sinwei, Shanghai, China) (Figure 2b). After drying at room temperature for 12 h, the sensors were trimmed and the undesired portions were removed. The sensors were then affixed to the actuator body using the aforementioned silicon adhesive (Figure 2f). To fit the finger-sized bellow-shaped actuator, the length, width, and thickness of the flexible bending sensor were selected to be 11 mm, 2 mm, and 1 mm, respectively. The length and width of the flexible bending sensors can be varied by changing the laser cutting pattern to match the dimensions of different soft actuators, while the thickness of the sensors can be adjusted by selecting a conductive sponge sheet with a different thickness. The number of sensors attached to the soft actuator depends on the required level of accuracy. In our design, we bonded nine thin sponge sensors to the actuator body.

### 2.2. Working Mechanism of Flexible Porous Piezoresistive Sensor

The resistance of our thin sponge sensor follows the relationship $R_{s}=C\left(l/A\right)$ since porous structures are evenly distributed inside, where *C* is a constant relative to the resistivity and porosity of the material. *l* and *A* represent the length and cross-sectional area of the sensor, respectively. The schematic diagram of the flexible sponge sensor and its operative principle is depicted in Figure 3. The left-hand side of the figure illustrates the fundamental components of the flexible sponge sensor. The electrical resistance of the sensor is sensitive to the angle of the attached object, as denoted by the variable α as shown on the right-hand side of Figure 3. By subjecting the attached object to external deformations, the angle α is likely to vary, thereby causing a change in the compression region of the sensor. This alteration in the compression region affects the electrical resistance of the sensor. Hence, the resistance of the flexible sponge sensor provides an indirect measure of the posture or deformation of the attached object. It is noteworthy that the object under investigation in this study is the bellow-shaped FEA. Developing a mathematical model for the sponge sensor would be inefficient due to the uncertain nature of its porous structure. Therefore, a learning-based method is more appropriate, and we utilized an LSTM neural network as our calibration neural network. The design of our LSTM neural network is presented in Section 2.4.

### 2.3. Soft Robot Kinematic Description

We utilized the parameter vector q=(k,ϕ,L) to represent the posture of the soft actuators (Figure 4), where *k* is the curvature of this bellow-shaped FEA. Here, we used a constant-curvature model to simplify the problem. Therefore, we have k=1/r, where *r* is the radius of the actuator arc. ϕ represents the angle between the x−z plane of the reference frame and the plane of the actuator arc. *L* represents the length of the actuator, which is relative to the length of the three parallel chambers $l_{1}$, $l_{2}$, and $l_{3}$. For our actuator, the length of each chamber $l_{i}$ (i=1,2,3) is relative to β, which is the angle between neighboring walls. For simplicity, we have $l_{i}=n\cdot a\cdot \sin\left(\beta /2\right)$, where *n* is the number of bellow units, and *a* is the length of the bellow wall segments.

In our case, we can directly measure the coordinates of the tip point $P_{1}=\left(x_{1},y_{1},z_{1}\right)$ and the rear point $P_{2}$ in the task space of the actuator. To transfer these coordinates to the configuration space, some manipulations are required. We set $P_{2}$ as the origin, and then ϕ can be calculated based on the tangent values of $x_{1}$ and $y_{1}$ if the projection of the actuator on the x−y plane does not lie on the primary axis.
(1)$$\begin{array}{c} \phi =\arctan\left(\frac{y_{1}}{x_{1}}\right) \end{array}$$

When the tip is located on the *z*-axis, the actuator is at singularity. In this case, we set ϕ to be zero.

Then, we rotate the actuator arc to the x−z plane by multiplying a rotational matrix about the *z*-axis. Then, we could obtain $P_{1}^{\prime }=\left(x_{1},y_{1},z_{1}\right)$ on the x−z plane,
(2)$$\begin{array}{c} \left[\begin{array}{c} x_{1}^{\prime }\\ y_{1}^{\prime }\\ z_{1}^{\prime } \end{array}\right]=\left[\begin{array}{ccc} \cos\left(-\phi \right) & -\sin\left(-\phi \right) & 0\\ \sin\left(-\phi \right) & \cos\left(-\phi \right) & 0\\ 0 & 0 & 1 \end{array}\right]\left[\begin{array}{c} x_{1}\\ y_{1}\\ z_{1} \end{array}\right] \end{array}$$

After that, we have
(3)$$\begin{array}{ccc} x_{1}^{\prime } & = & r(1-\cos\theta )\\ z_{1}^{\prime } & = & r\sin\theta \end{array}$$

Solving Equation (Equation 3), we can obtain the values of *r* and θ. Then, the actuator length *L* can be calculated using L=r·θ. Reference [44] provides more details on posture representations. In the rest of the paper, we will use either the parameter vector q or the tip point coordinates $P_{1}$ to represent the posture of the bellow-shaped FEA. We will use an LSTM neural network as a function approximator for the time-series response of sensors. The velocity, i.e., dynamic behavior, of the tip point of the actuator can be determined by taking the first-order differentiation of the tip coordinates, which is suitable for both the predicted value and the ground truth.

### 2.4. Neural Network Design

We employed an LSTM neural network to establish the relationship between the values of the thin sponge sensor and the posture of the bellow-shaped actuator. As previously discussed in Section 2.1, the development of an accurate mathematical model for the sponge sensor is an inefficient process. Hence, a learning-based approach was deemed optimal for our purposes.

The neural network architecture comprises an input layer, a fully connected layer, and an output layer, with the number of hidden layers ranging from one to four (validation results with different numbers of hidden layers are compared in Figure 5). Dropout layers follow the hidden layers to prevent overfitting. The inputs consist of a time-series sensing signal feedback from each sensor (R1, R2, R3…R9), and the output represents the absolute spatial position (x, y, z). The LSTM neural network’s structure is illustrated in Figure 6. The use of dropout layers helps to prevent overfitting. Furthermore, fine-tuning hyperparameters such as the number of hidden layers, number of hidden states, and dropout layer rate is critical to achieving optimal performance, which is compared and determined in Section 4.2. The overall root mean square error (RMSE) is selected as the cost function to evaluate the performance of the trained LSTM neural network, representing its ability to perceive the 3D dynamic posture of the bellow-shaped FEA in the presence of non-linear and hysteretic sensor feedback response. The MATLAB Deep Learning Toolbox is utilized to train and test the LSTM neural network, as detailed in Section 3.2.

## 3. Experiment Setup

### 3.1. Sensor Response Experiment Setup

In this experiment, we investigated the electrical resistance and rotating torque response of a thin sponge bending sensor as a function of the bending angle. One end of the sensor was affixed to a fixed plate, while the other end was attached to a rotational plate with a virtually fixed axis of rotation. The experimental setups for the electrical resistance and sensor torque response tests are illustrated in Figure A1a,b, respectively. In the experiment, to measure the electrical resistance response of the sensor, the rotational plate was driving by a worm gear and stepper motor, which allow the plate to rotate with a speed of 4${}^{\circ }$/s accurately. To mitigate the impact of fabrication defects, an enlarged sensor with dimensions of 30 mm in length, 5 mm in width, and 2 mm in thickness was employed for the resistance measurement test. The sensor was connected in series with a 220 Ω resistor in the measuring circuit. The value of angle β varied between 10${}^{\circ }$ and 100${}^{\circ }$ periodically for the diverse test. The electrical resistance of the sensor during the first ten cycles was recorded and calculated during the diverse cycle test. The average electrical resistance of the sensor when values of β are round numbers of ten are calculated with mean and standard deviation (S.D.) value. For the sensor torque testing setup, a torque sensor with a measuring range of ±0.1 Nm (SBT850A, Simbatouch, China) was positioned at the bottom of the rotational plate. Both the enlarged sensor and the sensor attached on the soft actuator were tested. The electrical conducting wires were removed to eliminate the effects of cable tension on the testing results. The torque was recorded once it reached a stable state. The sensor torque was measured in increments of 10${}^{\circ }$.

### 3.2. Soft Robot Operation and Data Acquisition

The experimental setup for soft robot operation and data acquisition included a vacuum pump that generated negative pressure, an MCU (Portenta H7, Arduino, Ivrea, Italy) equipped with three proportional valves (ITV2090-212L, SMC Corporation, Tokyo, Japan) to regulate the pressure inside each actuator chamber, a data acquisition card (USB-6212, National Instrument, Austin, USA), and a motion capture system (Bonita 10, Vicon Inc., Yarnton, UK).

To characterize the robot’s performance, the pressure inside each chamber was sequentially varied from 0 to −70 kPa at a rate of 2.44 kPa/s to explore the soft actuator’s working space. A comparison was made between soft actuators with and without sensors. During the data acquisition period for neural network training, the pressure within each actuator chamber was randomly varied within the range of 0 to −70 kPa with a 25 Hz refresh rate. Within each refreshing period, the pressure changed randomly from 0 to −70 kPa. Consequently, the pressure within each actuator exhibited a random triangular wave pattern.

In the data acquisition circuit, the resistance $R_{s}$ of the thin sponge sensors was converted into voltage information $V_{out}$ using voltage divider circuits with the relationship $V_{out}=V_{cc}R_{s}/\left(R_{0}+R_{s}\right)$, where $V_{cc}$ and $R_{0}$ are 0.5 V and 20 Ω, respectively. The sensor information $V_{out}$ was collected by a DAQ card at a frequency of 1000 Hz, which was sufficient for capturing the motion of pneumatic soft robots. During data processing, the sensor data was downsampled to a frequency of 100 Hz. A low-pass filter with a cut-off frequency of 20 Hz was applied to the sensor data to remove high-frequency noise.

The Vicon motion capturing system employed nine cameras distributed on the ceiling to capture the coordinates of reflective markers at a frequency of 100 Hz. The diameter and mass of the sphere markers were 1 cm and 0.42 g, respectively, and they did not significantly affect the performance of the soft actuator. One marker was placed on the top of the actuator to obtain the coordinates of the top point ($P_{1}$), while three markers were placed on the fixed rear part to obtain the coordinates of the rear point ($P_{2}$). The overall measuring error of the motion capturing system was less than 0.5 mm.

### 3.3. Neural Network Training

The data used to train the LSTM network were collected from random movements, with the pressure in the three chambers being randomly varied between 0 and −70 kPa. The training and validation data were obtained within 10 min and 400 s, respectively, with 70 s of validation data presented in Section 4.

The LSTM network training and generation were performed using the MATLAB Deep Learning Toolbox, with the default RMSE cost function and the Adam optimizer algorithm [45]. L2 regularization was adopted to prevent overfitting of the data. The initial learning rate was set to 0.001, and network performance was validated using a test set every 30 iterations, with overall iteration stopping after 100 epochs. The time-series data were sequentially divided, with 85% of the samples assigned to the training set and 15% assigned to the test set. The time series spanned 150 s and contained 9000 data points. Training the network with different sets of hyperparameters, including the number of hidden layers, number of hidden states, and dropout layer rate, was critical to optimizing network performance. These parameters were varied in the ranges [1, 2, 3, 4], [25, 50, 100, 200], and [0.1, 0.25, 0.5]. The root mean square error (RMSE) was used to evaluate the prediction accuracy, with $\mathrm{RMSE}=\sqrt{\mathrm{mean}\left[{\left({\mathrm{error}}^{*}\right)}^{2}\right]}$, where ${\mathrm{error}}^{*}=\sqrt{{\left(x_{pred}-x_{true}\right)}^{2}+{\left(y_{pred}-y_{true}\right)}^{2}+{\left(z_{pred}-z_{true}\right)}^{2}}$ represents the distance between the predicted and true actuator tip positions. The results of these calculations are summarized in Section 4.2. For the neural network with the lowest RMSE value, the neural network parameters were employed to estimate the data in the validation dataset, which is discussed in Section 4.3.

## 4. Experiment Results

### 4.1. Soft Sensor and Actuator Characterization

#### 4.1.1. Soft Actuator Working Space

In this experiment, the response characteristics of the soft actuator and sensors were explored, and the working space of the actuator with and without thin sponge bending sensors was determined. The tip point was treated as the origin in the experiment to facilitate comparison. When the pressure inside the three air chambers was varied from 0 to −70 kPa, the bending angle θ (as shown in Figure 4) varied between 0 and 90.69 degrees, while the actuator length *L* varied between 80.2 and 64.2 mm. Figure 7 shows the coordinates of the actuator tip point during changes in air pressure inside the chambers. The red curve represents the trajectory of the tip point when sponge bending sensors are attached to the actuator, while the blue curve represents the tip point trajectory when no sponge bending sensor is attached. These two working spaces are almost identical. Therefore, we can conclude that the attachment of distributed sponge bending sensors does not significantly affect the working space of the soft actuator.

#### 4.1.2. Response of Flexible Bending Sensor

The electrical resistance response of the sensor is presented in Figure 8a. The small cycles on the figure represent the mean value of the electrical resistance during the ten cycles test when the value of angle β is the round numbers of ten. The error bars are representing the correspondence standard deviation (S.D.) value. The resistance of the enlarged sponge bending sensor varied from 8 to 31 Ω as the angle between two rigid plates β changed from 10 to 100 degrees. The sensor’s resistance in the ten-cycle diverse test would go up gradually (Figure 9). Calibrating such sensor drifting using the conventional model-based calibration method would be challenging. In this work, by combining the flexible sensors and LSTM neural network calibration method, we can overcome the sensor drifting issue and predict the posture of the soft actuator with acceptable accuracy, which is presented Section 4.2. The sensitive region of the sponge bending sensor with these dimensions was found to be between 10 and 60 degrees. When the angle β exceeded 60 degrees, the sensor became less sensitive, while when the angle β was less than 10 degrees, most parts of the sensor were fully compressed, leading to saturation, as also shown in Figure 10. The sensitive region of the sponge bending sensor may vary depending on the sensor dimensions, but it can typically be divided into the aforementioned three sections.

As discussed in Section 3.2, the response of the flexible bending sensor when attached to soft pneumatic actuators was recorded, and the resulting voltage changes were plotted in Figure 10. Sensors #1–3, #4–6, and #7–9 were attached to the same bellow side wall from bottom to top, as shown in Figure 10a, resulting in similar responses of the three sensors with respect to actuator shape-changing. In this experiment, we determined the saturation region of the thin sponge bending sensor. When the air pressure was lower than −50 kPa, the voltage change due to actuator deformation was small and smooth (Figure 10b). Beyond this range, the sensor output saturated as the sponge edges fully contacted each other. This phenomenon is further discussed in Section 4.3.

Figure 8b depicts the sensor torque response of the sensor as a function of the sensor angle β. The torque exhibits an inverse relationship with respect to the angle β, with the blue and red lines representing the response of the sensor during angular decreases and increases, respectively. The maximum torque values for the enlarged sensor and the sensor attached to the actuator were found to be 16 mNm and 4.7 mNm, respectively. These results support the notion that the attached sensor does not significantly affect the working range of the bellow-shaped actuator, as discussed in Section 4.1.1.

### 4.2. Calibration Neural Network Performance

As delineated in Section 3.3, the process of optimizing the LSTM calibration neural network entailed fine-tuning three key parameters: the quantity of hidden layers, the number of hidden neurons, and the dropout rate in dropout layers. To gauge the performance of these neural networks, root mean square error (RMSE) values corresponding to the predicted tip point coordinates were computed, and the results are presented in Figure 5. An analysis of Figure 5 reveals that the LSTM neural network configuration with a single hidden layer, 100 hidden neurons, and a dropout rate of 0.5 yields the most favorable RMSE value of 12.57 mm. Subsequently, this LSTM neural network will be utilized to validate the 3D posture prediction of the soft actuator.

### 4.3. Validation of 3D Posture and Dynamic Behavior Prediction

Using the trained LSTM neural network discussed in Section 4.2, we were able to predict the 3D postures and dynamic behaviors of soft actuators, as demonstrated in Figure 11. The validation results presented in Figure 11a show three moments captured with time steps in the actuator random motions. Figure 11b presents 3D representations of actuator postures plotted with prediction and ground truth. For each moment, we calculated and listed parameter **q** based on the algorithm described in Section 2.3. Throughout the validation period, the predicted and ground truth displacements of the tip points are shown in Figure 11c, with displacements in the x-, y-, and z-axis directions represented by blue, red, and green colors, respectively. The dashed lines represent the ground truth of displacement in each direction. The RMSE values of the predicted value in the x-, y-, and z-axis directions were found to be 7.37 mm, 6.35 mm, and 3.70 mm, respectively. To assess the accuracy of the prediction, we evaluated the mean value and standard deviation (SD) value of prediction error in the coordinate axis direction, as reported in Table 1. The error percentage was computed by the absolute values of mean prediction error value divided by the working space of the actuator in the x-,y-, and z- axis direction. It is worth noting that the primary source of error was the saturation region, which can occur when the bending angle of the soft actuator is too large, causing the sensor to enter the saturation region. In our study, we found that a minimum of three sensors was required to estimate the robot’s posture. We present the posture estimation results for sensors arranged at low (sensors #1, #4, and #7), middle (sensors #2, #5, and #8), and high (sensors #3, #6, and #9) levels (shown in Figure 10) in Table 2.

By computing the differentiation of the coordinates of the tip points, we were able to obtain the linear velocity of the actuator tip, which characterizes the dynamic behavior of the soft actuator. Figure 12 illustrates the predicted tip velocities in the x-, y-, and z-axis directions compared to the ground truth, demonstrating the ability of our method to capture the dynamic behavior of soft actuators. The RMSE values for the predicted tip speeds in the x-, y-, and z-axis directions were found to be 3.14 mm/s, 3.17 mm/s, and 1.44 mm/s, respectively. To evaluate the accuracy of the speed prediction, we computed the mean and standard deviation of the prediction errors, which are reported in Table 3.

## 5. Conclusions and Discussion

This paper presents a novel end-to-end method for perceiving the 3D posture and dynamic behaviors of bellow-shaped soft actuators using distributed miniature flexible porous piezoresistive sensors. Additionally, we demonstrate the calibration of non-linear, hysteresis sensors using a trained RNN neural network. The scalable design of the sensor can be achieved by selecting a conductive sponge sheet with different thicknesses and laser-cutting patterns, making it applicable to different scales of soft robots. The porous structure and flexible material of the sensor minimize the influence on actuator motions, providing an advantage over previously developed flexible sensors, particularly in small-scale applications. A comparison between our flexible sponge sensor and recent typical works where RNN was applied in flexible sensor calibration is presented in Table 4. In comparison with the sensors developed in [38,46,47], we separated the design and fabrication of sensors from that of the attached actuator, thereby reducing fabrication difficulties and increasing design flexibility. Our flexible sponge sensor outperforms the sensors developed in [19] in capturing the velocity information due to the low inertia of the sponge material. Furthermore, compared with our previous work in [36,39], we have increased the degree-of-freedom of the motion.

### 5.1. Discussion on Kinematic Description of Soft Actuator

Although the posture of the soft actuator in this study was estimated based on the constant curvature concept, our end-to-end posture-perceiving method can be expanded to non-constant curvature situations. For soft actuators, this method is also suitable for estimating posture based on piecewise constant curvature (PCC), Bezier curves, or splines. The number and position of the sensor depend on the accuracy of the single sensor and the model employed to estimate the actuator posture, respectively. In this study, redundant sensors (theoretically only one sensor on each sub-chamber is necessary) were employed to improve the overall accuracy. For the piecewise constant curvature situation, each divided segment can be treated as having constant curvature, and the procedures in this study can be followed. For Bezier curves/splines situations, the sensor can be combined with the Bezier/spline model that enables non-constant curvature measurement. The neural network-based method can be used to find the parameters in the models or together with the model. The joint angle can be measured based on the method in this study, after which the joint angle distribution can be estimated. Finally, the overall posture of the soft actuator can be estimated.

### 5.2. Discussion on Sensor Characterization and Performance of the Sensing Method

As discussed in Section 4.1.2, both the sensitivity and resistive torque of the flexible sponge sensor exhibit a negative correlation with the angle β. Based on the results of ten diverse cycles of testing, it was observed that the electrical resistance of the flexible sponge sensor exhibited variations across different cycles. This issue poses a challenge for conventional model-based sensor calibration methods. However, the LSTM calibration neural network proposed in this paper was found to effectively address this challenge, enabling accurate calibration of the sensor and estimation of the posture of the soft actuators. It is worth noting that a saturation region arises when the sensor is consistently pressured. Specifically, when the sensors are adhered to the ravine of the bellow, the saturation occurs when the bellow is fully compressed by negative pressure. In the context of the present study, this saturation is reflected in the 3-DoF actuator under investigation when the actuator is fully compressed (i.e., all three bellows are fully compressed) or the bending angle θ is substantial (i.e., one or two bellows are fully compressed). Consequently, under these two conditions, the estimation error of the sensing system would be more pronounced, as evidenced by the data at 45 s when the bellow actuator is in random motion (see Figure 11). It is worth noting that this issue cannot be resolved by employing a calibration neural network, as there is an absence of effective training data.

The performance of velocity prediction is closely linked to the quality of the base material of the sensor, particularly its bandwidth with respect to mechanical vibration. With a larger bandwidth, the sensors can react to faster deformations of the attached objects. In the present study, we used an off-the-shelf conductive sponge whose base material is PU sponge, which has a slow recovery time to its original shape when pressure is removed. Consequently, when tracking the velocity of the actuator, the estimation error is substantial when the acceleration is large (reflected in Figure 12). Fortunately, for soft actuators, the response is typically slow.

During testing, it was observed that the flexible sponge sensor may experience fatigue after prolonged use, which is possibly due to damage of the conductive coatings. In the fabrication process of the conductive PU sponge, the base PU sponge was merged with a conductive solution. After drying, the internal fibers of the sponge were enveloped by a layer of conductive coating. During the use of the sensors, the coating may come into contact with neighboring coatings, leading to damage after prolonged use. Another issue pertains to the operating temperature, as the PU material is sensitive to high-temperature environments. The heat generated by friction during operation may also shorten the sensor’s lifetime. The presence of noise resulting from bare enameled wire during sensor resistance measurement may also have an impact on the predictive accuracy of the calibration neural network.

Moving forward, we aim to enhance the sensitivity, accuracy, and longevity of the sensor. Moreover, we plan to investigate additional applications of the proposed sensing system within the realm of soft robotics.

## Figures and Tables

**Figure 1 sensors-23-06189-f001:**
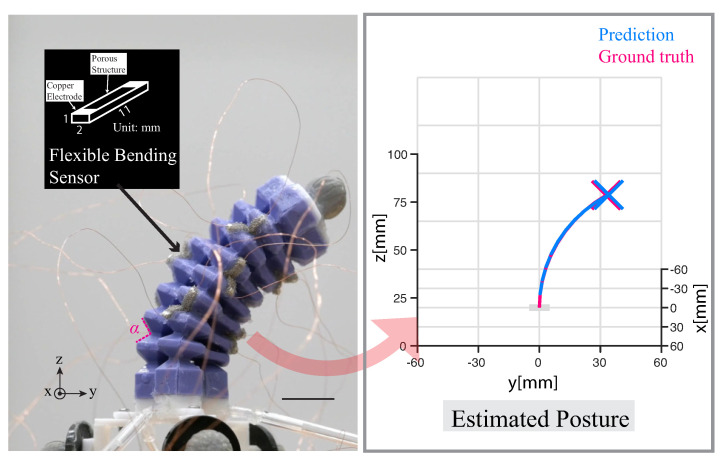
Proprioception of posture in a pneumatic soft actuator. On the side of the actuator (left), there are nine distributed flexible bending sensors (scale bar = 1 cm) made by a thin conductive sponge via the planar fabrication method. The neighbor wall angle α was indicated. A trained neural network was employed to calibrate the sensor, and the 3D representation of the posture of soft actuator is shown on the right. (The perspective of the right figure is a top–down view along the x − z plane, parallel to the vector (3, 0, 1)).

**Figure 2 sensors-23-06189-f002:**
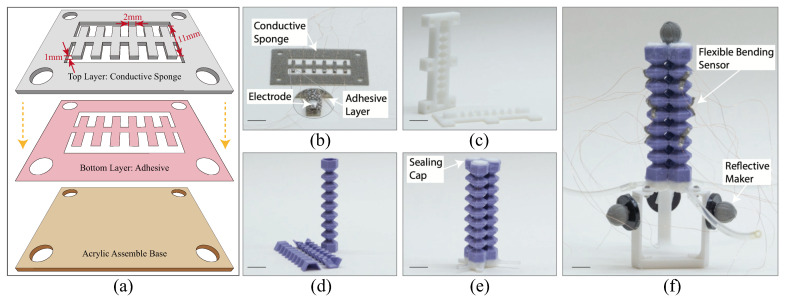
Fabrication of flexible bending sensor and pneumatic soft actuator. (**a**) Sketches of planer fabrication process of sensor body. (**b**) Flexible bending sensor via planar fabrication method. (**c**) Molds for silicone casting. (**d**) Half-chamber of the soft actuator made by silicon after the demolding process. Two half-chambers were combined to form one air chamber using silicon adhesive. (**e**) Three air chambers combined via silicon adhesive with top and bottom caps. (**f**) Soft pneumatic actuator with distributed flexible bending sensors. Sensors were attached to the actuator side via silicon adhesive. Four reflective makers are used to determine the true posture of the soft actuator (scale bar = 1 cm).

**Figure 3 sensors-23-06189-f003:**
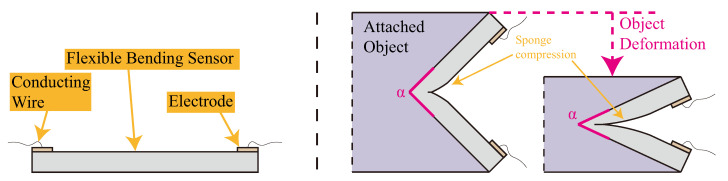
The cross-view sketch in the left panel shows the flexible sponge sensor, while its working mechanism is depicted in the right panel. When there is deformation on the attached object, the sponge compression region changes, resulting in a change in electrical resistance.

**Figure 4 sensors-23-06189-f004:**
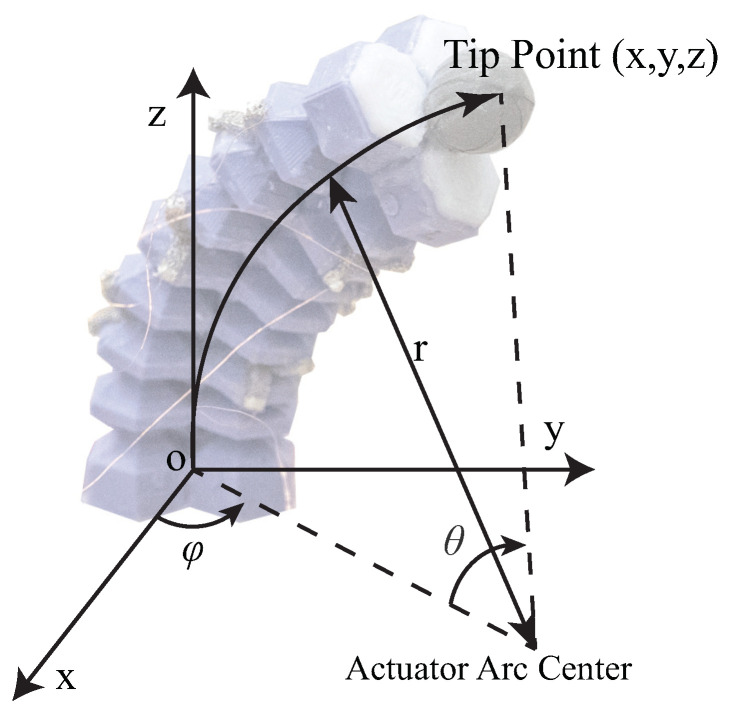
Kinematic representation of soft pneumatic actuator. The principle axis of reference frames was indicated.

**Figure 5 sensors-23-06189-f005:**
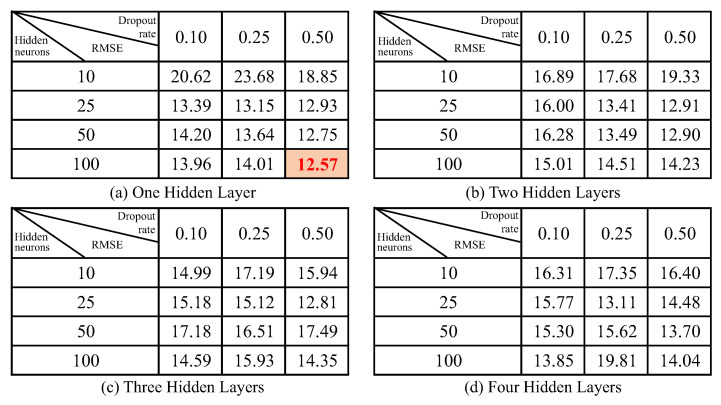
RMSE of validation set prediction for LSTM neural network with different neural network parameters (unit: mm).

**Figure 6 sensors-23-06189-f006:**
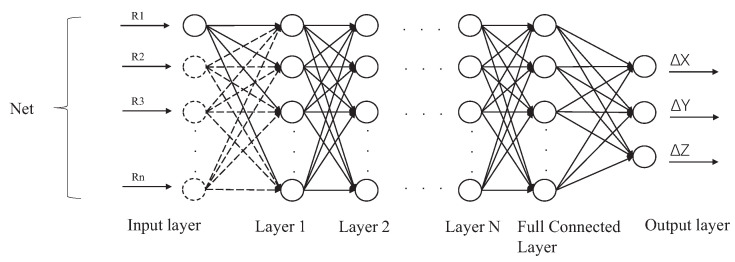
Structure of LSTM neural network. The neural network contains the input layer, hidden layer, fully connected layer, and output layer.

**Figure 7 sensors-23-06189-f007:**
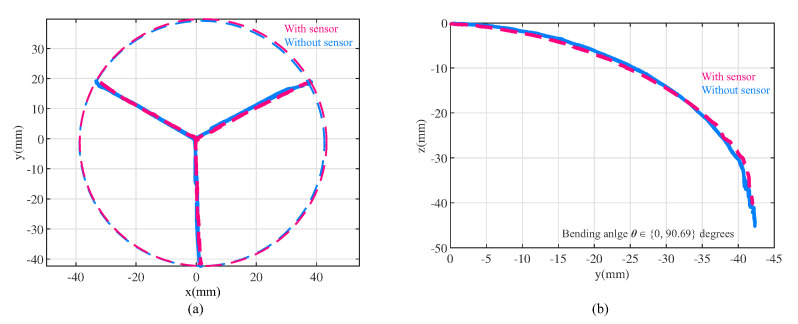
Working space of bellow−shaped actuator with (red curve) and without (blue curve) sponge bending sensor. (**a**) Top view of working space. (**b**) Side view of working space. The bending angle θ of the actuator could vary between 0 and 90.69 degrees. This shows that the thin sponge sensor has negligible influence on the soft actuator’s motion.

**Figure 8 sensors-23-06189-f008:**
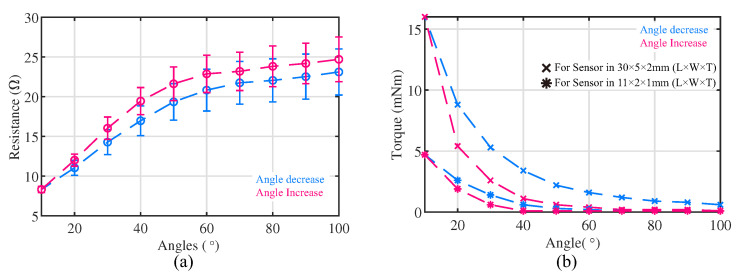
Experimental results of flexible bending sensor in (**a**) electrical resistance response test and (**b**) sensor torque response test. The responses of sensors when the bending angle decreased and increased are indicated by the blue line and red line, respectively.

**Figure 9 sensors-23-06189-f009:**
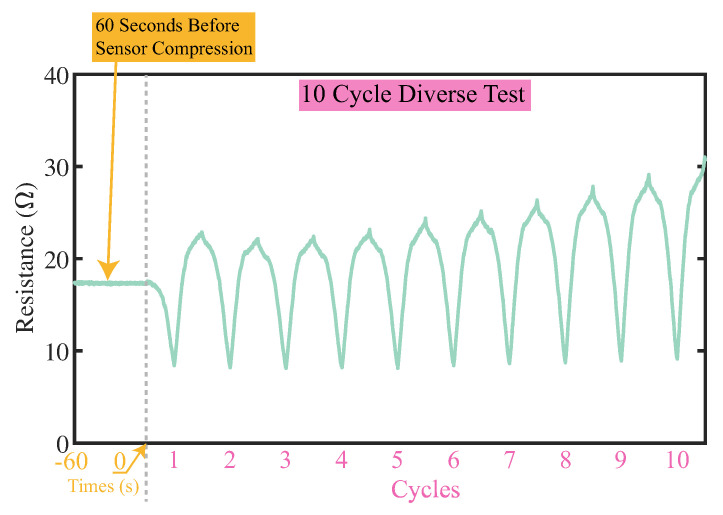
Diverse test results. Resistance of sensor during the 10−cycle diverse test together with 60 s data before test are presented to indicate sensor responses.

**Figure 10 sensors-23-06189-f010:**
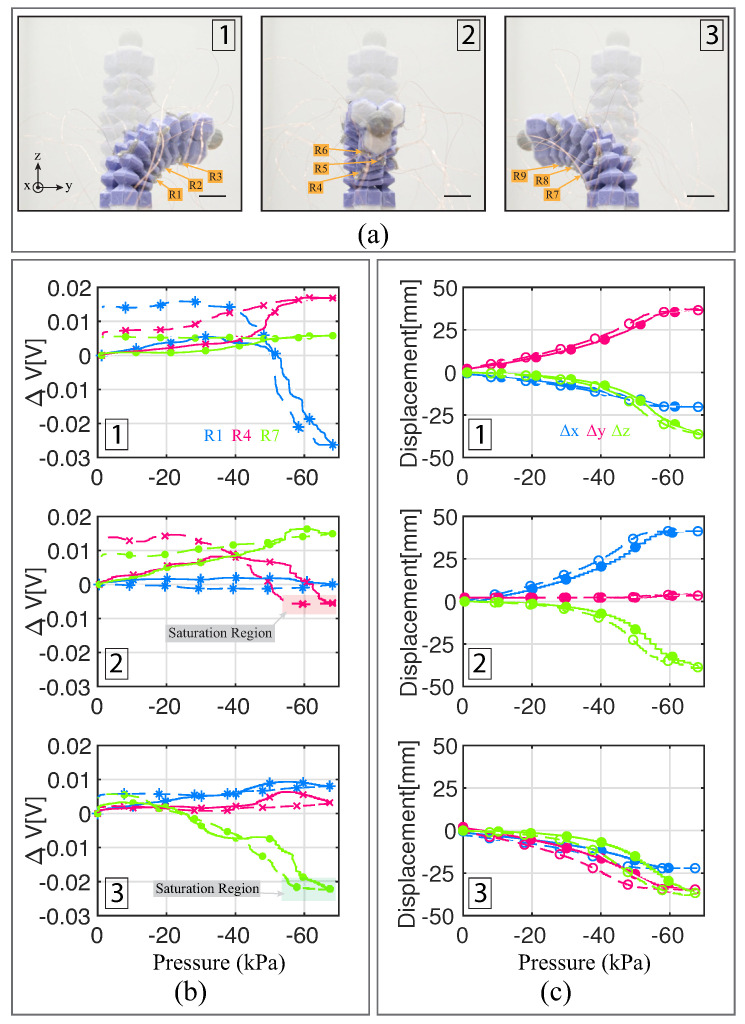
Characterization of thin sponge bending sensor integrated pneumatic soft actuator. (**a**) Photos show that −70 kPa negative pressure was applied to air chambers 1, 2, and 3, respectively (scale bars = 1 cm). (**b**) Sponge bending sensor (#1, #4 and #7) response (voltage changing) with respect to pressure changing in three air chambers with an interval of 1 kPa. (**c**) Tip displacement of actuator versus deflation pressure applied to each air chamber with same actuation pattern shown in (**a**). The tip point was treated as the original point. Index 1, 2, and 3 in each figure indicating each actuated air chamber. In both (**b**) and (**c**), filled lines and dashed lines are representing bending and relaxing processes, respectively. A median filter was applied in the plotting of (**b**).

**Figure 11 sensors-23-06189-f011:**
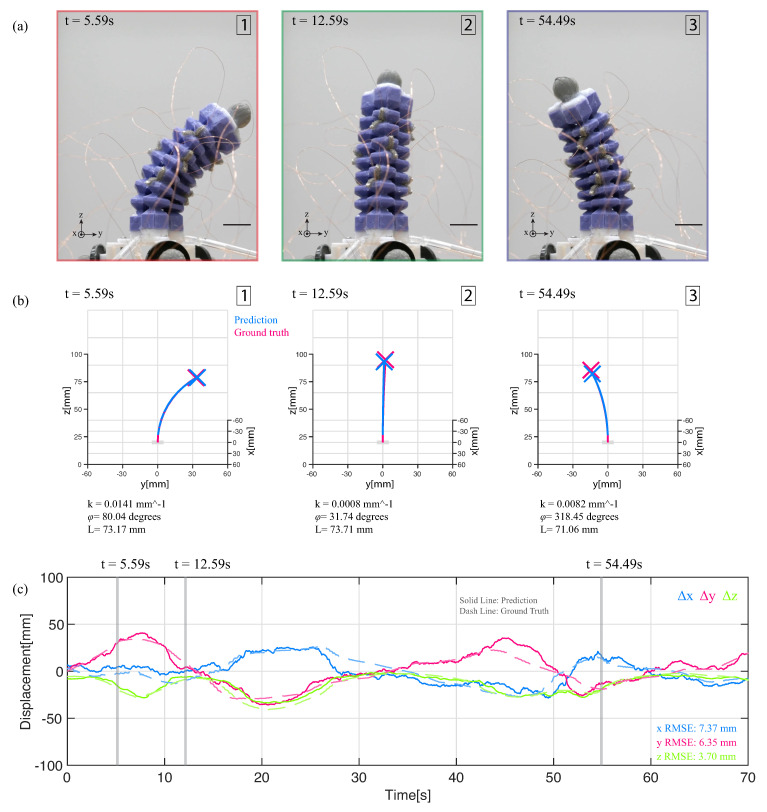
Validation results on random actuator motions. (**a**) Photos with time step. (**b**) Respective 3D representations of postures of the soft actuator. Kinematic parameters *k*, ϕ, and *L* are shown in the figure. In (**b**), the blue curve represents predictive posture; the red curve represents ground truth. (**c**) Plots of ground truth (dashed lines) and predictive displacement in the x−axis direction (blue curve), y−axis direction (red curve), and z−axis direction (green curve) of the reference frame. Dashed lines are representing ground truth. Time steps in (**a**,**b**) are represented by a gray shaded vertical bar (scale bar = 1 cm. The perspectives of (**b**) are top−down view along the x − z plane, parallel to the vector (3, 0, 1)). Actuator motions in (**c**) are included in Appendix A.

**Figure 12 sensors-23-06189-f012:**
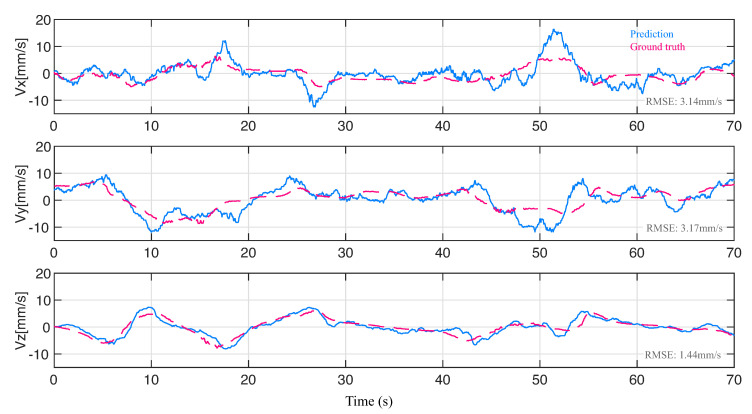
Velocity prediction of actuator tip for validation results in Figure 11c. From top to bottom, velocity in the prime axis direction of the reference frame was represented. Blue curves represent predictive velocity; red dash lines represent ground truth.

**Table 1 sensors-23-06189-t001:** Prediction accuracy of actuator tip position for validation data set.

Direction	Prediction Accuracy (mm)	Error Percentage
x-axis	−0.31 ± 7.37	0.37%
y-axis	2.00 ± 6.03	2.38%
z-axis	1.33 ± 3.46	1.58%

**Table 2 sensors-23-06189-t002:** Prediction accuracy of actuator tip position for validation data set using single-layer thin sponge flexible sensors.

Sensors	Prediction Accuracy in x-Axis (mm)	Prediction Accuracy in y-Axis (mm)	Prediction Accuracy in z-Axis (mm)
#1, #4 and #7	3.21 ± 11.82	0.43 ± 9.53	1.78 ± 5.47
#2, #5 and #8	−5.19 ± 11.41	4.39 ± 14.38	6.91 ± 7.05
#3, #6 and #9	−2.46 ± 9.99	−4.97 ± 9.39	4.83 ± 6.62

**Table 3 sensors-23-06189-t003:** Prediction accuracy of actuator tip speed for validation data set.

Direction	Prediction Accuracy (mm/s)
x-axis	0.48 ± 3.11
y-axis	−0.25 ± 3.16
z-axis	0.11 ± 1.44

**Table 4 sensors-23-06189-t004:** Comparison among our work and recent typical works where RNN was applied in flexible sensor calibration.

	Research Work	Sensor Material /Structure	Design Method	Advantages	Drawbacks
#1	Truby et al. [19]	Conductive Silicon	Separated	3-DoF Posture Estimation	Only Steady-State Estimation
#2	Shu et al. [39]		Separated	Dynamic Posture Estimation	Only Performs on Single DoF Motion
#3	Thuruthel et al. [38]		Integrated	2-DoF Posture and Contact Force Estimation	Low Sampling Frequency (10 Hz)
#4	Truby et al. [46]	Sealed Air Chamber	Integrated	3-DoF Dynamic Posture Estimation and Tactile Detection	Require Specific Fabrication Equipment (High-Resolution DLP 3D Printers)
#5	Ang et al. [47]		Integrated	2-DoF Dynamic Posture and Contact Force Estimation	Low Sampling Frequency
#7	Shu et al. (This Work)	Conductive Sponge	Separated	3-DoF Dynamic Posture Estimation	Saturation Region Affects Estimation Accuracy

## Data Availability

Not applicable.

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
