# Peer review of "An End-to-End Dynamic Posture Perception Method for Soft Actuators Based on Distributed Thin Flexible Porous Piezoresistive Sensors"

_sensors, 2023, doi:10.3390/s23136189_

Round 1
Reviewer 1 Report
Well-written and detailed work on soft actuators with piezo-resistive senors, good quality work. I only have minor comments below:
Figure 5a could be more clear. For example, the sample could be indicated the and wires for electrical resistance could be indicated as they are difficult to see.
Figure 6 - no need for square box around whole graphs.
Figure 7- no need for titles at the top of graphs as you have figure heading below, provides a cleaner figure.
A recent review that is potentially of interest to soft robotics.
Soft Actuators and Robotic Devices for Rehabilitation and Assistance
M Pan et al Advanced Intelligent Systems 4 (4), 2100140 2022
Generally good but some minor changes advised with careful read through.
Reviewer 2 Report
In this manuscript, the author employs an array of 2 miniaturized sponge resistive materials along the soft actuator, which uses long short-term memory 3 (LSTM) neural networks to solve the end-to-end 3D posture for the soft actuators. The effectiveness of the method has been demonstrated on a finger-size 3 degree of 8 freedom (DOF) pneumatic bellow-shaped actuator, with nine flexible sponge resistive transducers 9 placed on the soft actuator’s outer surface. This review is clear for understanding the subject. This work is recommended to be published before the following observations are addressed.
1) Design of the sensor is not clear.
2) Diverse cycle tests of the sensor should be performed.
3) Error bars are recommended to be added in Figure 7.
4) Graphs for experimental setups can be moved to the supporting information.
Title may be slightly modified. “Transducers” to be replaced by “sensors”?
5) For robots’ posture detections, a number of recently important paper have not been added. The authors can refer to these publications. The authors can refer to these publications. 1. Advanced Sensor Research, 2023: 2300044; 2. Advanced Materials, 2021, 33(18): 2008701; 3. Advanced Functional Materials, 2021, 31(39): 2007436. 4. Journal of Materials Chemistry C, 2019, 7(31): 9609-9617.
NA
Reviewer 3 Report
This manuscript reports a method for an accurate 3D posture of the soft actuators using closed-loop control of soft robots. This method employs sponge-resistive elements along the soft actuator that uses long short-term memory (LSTM) neural networks to improve the end-to-end 3D posture of soft actuators. However, this manuscript must be improved based on the following issues:
1.-The abstract should add experimental results of the proposed method.
2.-The second graph of Figure 1 (estimated posture) should be drawn in 3D, including the three axes (x,y,z).
3.-The introduction should consider the limitations of the proposed method.
4.-This manuscript requires a table that includes the proposed method’s main advantages, drawbacks, and operating parameters compared with others reported in the technical literature.
5.- The second section (Design overview and rationale) should incorporate more information on the description of the different stages of the proposed sensor design. In addition, this section could add more views of the sensors’ dimensions, materials, performance parameters, and positions on the soft actuator. Also, the description of the operating principle and electrical connections of the bending sensor and pneumatic soft actuator can be enhanced.
6.- The fourth section (experimental results) should incorporate discussions on the noise in the measurements. Furthermore, the authors could consider discussions on the environmental temperature’s effect on the proposed sensor’s performance.
7.-The discussions of the results shown in Figures 7, 8, 9, and 10 must be improved.
8.- Figure 9b should be drawn in 3D.
9.- The authors should include more experimental results of the actuator motions.
10.- The positions of the Tables and Figures must be revised.
The English grammar is acceptable.
Round 2
Reviewer 3 Report
This manuscript was enhanced based on the reviewer's comments.
The English can be improved.